# Experimental Carbonation Study for a Durability Assessment of Novel Cementitious Materials

**DOI:** 10.3390/ma14216253

**Published:** 2021-10-21

**Authors:** Lucija Hanžič, Sebastijan Robič, Alisa Machner, Marie Helene Bjørndal, Klaartje De Weerdt, Yushan Gu, Benoît Bary, Rosa Maria Lample Carreras, Aljoša Šajna

**Affiliations:** 1Laboratorij za Beton, Zavod za Gradbeništvo, ZAG, Dimičeva Ulica 12, SI-1000 Ljubljana, Slovenia; sebastijan.robic@zag.si (S.R.); aljosa.sajna@zag.si (A.Š.); 2Centrum Baustoffe und Materialprüfung, Technische Universität München, TUM, Franz-Langinger-Str. 10, 81245 Munich, Germany; alisa.machner@tum.de; 3Institutt for Konstruksjonsteknikk, Norges Teknisk-Naturvitenskaplige Universitet, NTNU, Richard Birkelandsvei 1A, 7491 Trondheim, Norway; marie.bjorndal@norconsult.com (M.H.B.); klaartje.d.weerdt@ntnu.no (K.D.W.); 4Service d’Étude du Comportement des Radionucléides, Université Paris-Saclay, CEA, 91191 Gif-sur-Yvette, France; yushanjoanna@hotmail.com (Y.G.); benoit.bary@cea.fr (B.B.); 5Acciona Construction, 28108 Alcobendas, Spain; rosamaria.lample.carreras@acciona.es

**Keywords:** mortar, absorption of water, carbonation, durability assessment, model verification

## Abstract

Durability predictions of concrete structures are derived from experience-based requirements and descriptive exposure classes. To support durability predictions, a numerical model related to the carbonation resistance of concrete was developed. The model couples the rate of carbonation with the drying rate. This paper presents the accelerated carbonation and moisture transport experiments performed to calibrate and verify the numerical model. They were conducted on mortars with a water-cement ratio of either 0.6 or 0.5, incorporating either a novel cement CEM II/C (S-LL) (EnM group) or commercially available CEM II/A-S cement (RefM group). The carbonation rate was determined by visual assessment and thermogravimetric analysis (TGA). Moisture transport experiments, consisting of drying and resaturation, utilized the gravimetric method. Higher carbonation rates expressed in mm/day^−0.5^ were found in the EnM group than in the RefM group. However, the TGA showed that the initial portlandite (CH) content was lower in the EnM than in the RefM, which could explain the difference in carbonation rates. The resaturation experiments indicate an increase in the suction porosity in the carbonated specimens compared to the non-carbonated specimens. The study concludes that low clinker content causes lower resistance to carbonation, since less CH is available in the surface layers; thus, the carbonation front progresses more rapidly towards the core.

## 1. Introduction

The design process of concrete structures relies on standards such as Eurocode 2 [1,2] and guidelines such as fib Model Code 2010 [3]. The main focus of the existing codes and guidelines is on mechanical performance, which is analyzed with advanced structural design models. In contrast, durability-related phenomena are addressed mainly through the selection of exposure classes and experience-based requirements. As a result, concrete durability forecasts are only crude approximations, which are not supported by the sophisticated modelling of concrete degradation over time. The deterioration and maintenance of concrete are having a significant impact on public sector budgets. Maintenance costs and shutdowns of infrastructure, such as tunnels, bridges or power plants, are particularly important, due to their impact on the wider community. 

When concrete is exposed to an aggressive environment with an increased concentration of carbon dioxide in combination with moisture, concrete undergoes a carbonation process. The main process in the carbonation of concrete is the reaction of portlandite, that is, calcium hydroxide Ca(OH)_2_, commonly denoted CH, with carbon dioxide (CO_2_). The reaction products are calcite, that is, calcium carbonate (CaCO_3_), and water (H_2_O). This reaction leads to a net mass gain, as the uptake of CO_2_ (44 g mol^−1^) leads to a larger mass increase compared to the release of water (12 g mol^−1^). While the products of this chemical reaction are not detrimental to concrete durability per se, the consumption of CH reduces the pH value of the pore solution, which may lead to the corrosion of reinforcement steel. The rate of carbonation is influenced by the curing conditions of concrete and by the climatic and local conditions to which the concrete is exposed. To accurately predict concrete deterioration, a numerical model for concrete carbonation is being developed [4] within the Horizon 2020 project, titled EnDurCrete [5]. The model is based on previous work conducted by Bary et al. [6,7,8]. It couples the mass conservation equations of water, CO_2_ and calcium in pore solution with thermodynamic modelling, which simulates the phase assemblage of binders upon carbonation. The model is used for the prediction of concrete behavior under accelerated carbonation, particularly where novel, more sustainable cements are used.

The European Commission and the European construction industry push to improve and develop technologies that reduce CO_2_ emissions as well as energy and material usage in the cement production processes [9]. Many studies attempt to develop environmentally-friendly concrete by reducing Portland clinker content in cement and substituting it with supplementary cementitious materials. In this manner, the above-mentioned model development is also linked to another aim of the EnDurCrete project [5], namely, the development of low carbon-footprint binders aligned with the new edition of prEN 197-1 [10]. One such novel cement is CEM II/C (S-LL), a low-clinker cement containing high-value industrial by-products. The use of this cement is aimed at the creation of cost-effective sustainable concrete on one hand, and high durability concrete, which can withstand exposure to aggressive environments, on the other hand. Details of the novel cement CEM II/C (S-LL) developed by HeidelbergCement are summarized by Bolte et al. [11].

This study aims to present the experimental results of an accelerated carbonation program conducted on mortar specimens incorporating a novel CEM II/C (S-LL) binder developed within the EnDurCrete project [5], and to compare it with the performance of mortars prepared with commercially available cement, CEM II/A-S. The specimens underwent accelerated carbonation in a conditioning chamber with controlled CO_2_ concentration, temperature, and relative humidity. The carbonation rate was assessed by measuring the depth of carbonation over time, (i) visually, using a pH indicator, and (ii) by thermogravimetric analysis (TGA). The impact of carbonation on moisture transport was evaluated by the gravimetric method on specimens exposed to a specific protocol of drying and wetting.

The results of the accelerated carbonation tests will serve for the calibration and validation of the numerical model dealing with carbonation processes in concrete, in particular, the coupling between saturation degree, water transport and carbonation extent. The model itself is out of the scope of this paper. Its detailed description, parameter calibration procedure and results will be comprehensively presented in a separate publication. In addition, the results aim to further the understanding of concrete deterioration in a CO_2_-rich environment, particularly for concrete made with novel eco-friendly cements. Consequently, the design of concrete structures in terms of durability will be improved, resulting in reduced maintenance costs and shutdowns of infrastructure caused by the deterioration of concrete due to carbonation.

## 2. Materials and Methods

The concept of the experiment is derived from the requirements of the carbonation model, where two coupled mass balance equations are considered, one covering the transport of moisture and the other the migration of CO_2_ in the interconnected pores. The carbonation process is simplified into the dissolution of a combined Ca-containing hydrate phase (mainly CH and C-S-H) and the precipitation of calcite according to thermodynamic modelling. The evolution of their volumes is approximated as a piecewise linear function. The dissolution rate of the Ca-containing hydrate phase is assumed to depend on the saturation degree, the partial pressure of CO_2_, and the fraction of the current volume versus the initial volume, from which the precipitation rate of calcite can be predicted.

For these reasons, this experimental study encompassed two test procedures. The first was the determination of the carbonation rate, which included measurements of carbonation depth and portlandite profiles, which were subsequently used to verify the carbonation model and thus estimate the evolution of porosity, saturation degree, and phase changes. The second test procedure was a moisture transport experiment, from which the initial porosity and both the initial and relative permeability coefficient were quantified. Since carbonation leads to microstructure changes and then affects transport properties [12], tests before and after carbonation in different relative humidity conditions were carried out. To complement the moisture transport test, the total porosity was also measured.

### 2.1. Specimen Preparation

Concrete used in applications where carbonation resistance is required is normally designed with a relatively low water/cement (w/c) ratio and therefore measurable carbonation only occurs during long-term exposure. To accelerate the process, the mixes used in this study had a higher w/c ratio than is usually applied for such structures, namely 0.50 or 0.60, so that measurable carbonation took place in a reasonable timeframe. Furthermore, coarse aggregates were excluded to minimize the effects of the interfacial transition zone.

The mortar mix designs are shown in Table 1. The specimens were labelled EnM for the EnDurCrete mortar using novel CEM II/C (S-LL) cement, and RefM for the reference mortar using CEM II/A-S cement. The label extension -05 or -06 was used to denote w/c ratio.

The two cements were composite blends containing the same types of components, namely CEM I 52.5 R, ground granulated blast furnace slag (S), and limestone filler (LL). CEM I 52.5 R consists predominantly of clinker (K), whose content amounts to >95 wt%. There were, however, two important differences between the two blends. Firstly, the mass ratio of components was different, as shown in Table 2. The novel CEM II/C (S-LL) contained significantly less clinker, which was replaced by slag and limestone. The chemical composition of the two cements is given in Table A1 of the Appendix A. Secondly, the components of CEM II/C (S-LL) were ground separately to optimize packing and reactivity, as opposed to the reference cement CEM II/A-S, where the components were inter-ground. Details on cement development are provided by Bolte et al. [11]. The different composition and grinding procedures are reflected in the physical properties summarized in Table 2.

The aggregate used for the mortar mixes was limestone/quartz river sand with a maximum grain size of 4 mm. Its physical properties are summarized in Table 3, while the particle size distribution is presented graphically in Figure 1.

Along with the specimens for other research programs of the EnDurCRete project, three mortar prisms measuring 10 cm × 10 cm × 40 cm were cast per mix for this carbonation study. The processing of the prisms is graphically presented in the flowchart in Figure 2. The prisms were cured in a humidity chamber with relative humidity (RH) > 95% at 20 ± 2 °C for 21 days as per EN 12390-2 [13]. Next, the prisms were wrapped in plastic foil and transported from the casting to the testing facilities, where they were cured at 20 °C for up to 76 days. The moisture state of the specimens was checked on one prism at this stage using the method described in [14,15] to confirm that the specimens did not dry out.

After curing, the prisms were cut laterally either into thick blocks with dimensions 10 cm × 10 cm × 7 cm or into thin plates with dimensions 10 cm × 10 cm × 2 cm. The end sections were discarded. This way, either four thick blocks or eight thin plates were obtained from one prism. The thick blocks were used for the carbonation rate measurements, while the thin plates were used for the moisture transport experiments.

### 2.2. Carbonation Rate

Four thick blocks of each mix were placed into the carbonation chamber with a temperature of 21 ± 2 °C, RH 60 ± 10% and 1% CO_2_ according to EN 13295 [16]. One block was taken out of the carbonation chamber after 14, 28 and 90 days and, depending on the w/c ratio, either after 146 (w/c 0.60) or 167 days (w/c 0.50) of exposure. It was split perpendicularly to the sawn surface. The carbonation front was visually determined on one half with a ruler on a freshly split surface with dimensions 7 cm × 10 cm, which was sprayed with a thymolphthalein pH-indicator solution. Where the mortar was unaffected by the carbonation (pH above 9.3 to 10.5), the indicator turned blue, while the carbonated surface remained uncoloured. The results were plotted against the square root of time and the carbonation rate was calculated as the slope of the fitted line given by the following equation:(1)$$d_{k}=a+K_{\mathrm{AC}}\times \sqrt{t}$$
where *d*_k_ (mm) is the mean carbonation depth at time *t* (days), *a* (mm) is a constant representing the y-axis intercept, and *K*_AC_ (mm day^−0.5^) is the carbonation rate [17].

For the TGA method, the other halves of the split blocks were used. The halves were profile-ground inwards from the sawn surface. Approximately 30 g of powder extracted from the consecutive layers was analyzed using a Mettler Toledo TGA/DSC3+ device (Mettler-Toledo, Kowloon, Hong Kong, CHN) in a temperature range from 40 to 900 °C, at a heating rate of 10 °C min^−1^. During the measurements, the measurement cell was purged with 50 mL N_2_ per minute. The amount of CH relative to the ignited mass of the powder (*CH*_ignited_) at 900 °C was calculated as
(2)$$CH_{\mathrm{ignited}}=\frac{w_{400}-w_{550}}{w_{900}}\times \frac{M_{\mathrm{CH}}}{M_{H_{2}O}}$$
where $w_{400}$, $w_{550}$ and $w_{900}$ (g) refer to the mass of powder at 400, 550 and 900 °C respectively and *M* (g mol^−1^) is the molar mass of either CH or water [18]. The weight loss due to the decomposition of portlandite was determined with the integration method of the derivative thermogravimetric (DTG) curves described by Lothenbach et al. [19]. Since carbonation consumes CH, its content is a measure of the degree of carbonation. The CH content was plotted as a function of the depth from the exposed surface. A detailed description of the experimental setup can be found in [15].

### 2.3. Moisture Transport

The moisture transport experiment was carried out on the thin plates. It consisted of several stages of conditioning, as shown in Figure 3, where stage 1 corresponds to curing, as described in Figure 2 under the “Casting and curing” tab. After 76 days of curing, the thin plates were sawn and organized into two groups, namely group “0%” and group “3%”, indicating the CO_2_ concentration to which groups were exposed in stage 2. Throughout the experiment, the mass of the plates was measured at regular intervals using a balance with an accuracy of 0.01 g.

In stage 2, the thin plates of the “3%” group were placed in the carbonation chamber with a temperature of 21 ± 2 °C, RH 60 ± 10% and 3.1% CO_2_, according to EN 13295 [16]. The higher CO_2_ concentration, compared to the 1% used on the thick blocks in Section 2.2, was assumed to have no impact on the type of phases that form during the carbonation process. This assumption is based on findings by Revert et al. [20,21], who investigated the impact of accelerated carbonation on microstructure and phase assemblage and found that a concentration of up to 5% is representative of natural carbonation. The thin plates in the “0%” group were stored in the conditioning chamber, with a temperature of 18–25 °C and RH 50–65%, at a reduced CO_2_ concentration. Soda-lime as a CO_2_ trap was put under the specimens to capture carbon dioxide and reduce its concentration in the chamber’s atmosphere. Stage 2 lasted for 146 days. At the end of stage 2, one plate from each group was split in half and the carbonation depth was measured using a phenolphthalein pH-indicator solution. The rest of the plates proceeded to stage 3.

In stage 3, the remaining plates from both groups were re-saturated. The plates were first saturated by capillary absorption. The water level was gradually increased over several days until the specimens were completely immersed. Again, the mass of the plates was measured at regular intervals. When they all reached a near-to-constant mass, they were further saturated by vacuum saturation according to EN 12390-11 [22]. Stage 3 lasted for 63 days.

In stage 4, the specimens were kept in normal atmospheric conditions at 20 °C and RH 60%. This stage lasted 33 days. Once the plates reached a constant mass, they were further conditioned at 20 °C and RH 30% in stage 5.

Upon the completion of stage 5, the total porosity of the specimens was determined according to SIA 262-1 [23]. This is a gravimetric method in which the specimen is first weighed after drying at 50 °C, next, the specimen is submerged in water for several days and weighed, both when submerged in water and when above water. The specimen is dried again at 50 °C, vacuumed at pressure < 1 mbar and submerged in water and weighed again under the water and above the water. Finally, the specimen is weighed after being dried at 110 °C.

## 3. Results and Discussion

The results of the accelerated carbonation study, where the carbonation rate was measured by a pH-indicator and by TGA, are presented in Section 3.1. The results of both methods are compared and discussed and are found to have a good correlation. The moisture transport results, given in Section 3.2, indicate an increase in porosity due to carbonation in specimens with a w/c ratio of 0.6.

### 3.1. Carbonation Rate

The carbonation depth results assessed visually after spraying the pH-indicator solution on the freshly split surfaces are given in Figure 4, where carbonation depth is plotted against the square root of time. In general, there is a linear relationship between the carbonation depth and the square root of time, therefore a straight line (Equation (1)) was fitted to the results, adopting a confidence level of 95%. The results of the fitting process are summarized in Table 4. The R^2^ values ranged from 0.79 to 0.98, indicating a reasonably-good-to-excellent fit. In the two cases where the R^2^ values were less than 0.90, the residuals were evenly distributed.

The results thus show that the carbonation rate of ~1.0 mm day^−0.5^ found in the EnM-06 was the highest, while the RefM-05 had the lowest carbonation rate of ~0.2 mm day^−0.5^. The other two mortars, namely EnM-05 and Ref-06, had approximately the same carbonation rate of ~0.6 mm day^−0.5^. Consequently, after 146 days of exposure, the EnM-06 demonstrated a carbonation depth of 13 mm as opposed to the 9 mm measured on the RefM-06. Similarly, the EnM-05 demonstrated a carbonation depth of 8 mm while only 4 mm was measured on RefM-05. The results thus indicate that a higher w/c ratio leads to a higher carbonation depth. This was expected, since a higher w/c ratio generates higher porosity of the cement matrix, allowing a faster penetration of CO_2_ into the sample [24,25,26,27]. 

The CH content determined by TGA as a function of depth for the samples prepared with a w/c ratio of 0.6 is shown in Figure 5. Selected individual TGA results are presented in Figure A1 and Figure A2 of the Appendix A. In general, the CH content decreased towards the surface, where no CH could be detected because all of it was consumed during the carbonation reaction. The plateau level of the CH content determined on the deeper sections represents the amount initially present in the uncarbonated mortars. These results show that CH content in mortar made with the novel cement with low clinker content (EnM-06) was about 1.5 wt% compared to the 3 wt% CH present in mortar made with the reference cement (RefM-06). Since less CH is present in the EnM-06, the carbonation front was found further from the top surface; for example, after 90 days of carbonation it was found ~12 mm from the surface, while in RefM-06 it was found at ~8 mm. A similar occurrence was observed by Carneiro et al. [28], who used specimens with low and high Portland cement content and found the carbonation front significantly deeper in specimens with low cement content. The depth of carbonation front measured with TGA corresponds well to the carbonation depth measured with the visual method and, therefore, with the results previously shown in Figure 4. 

The lower carbonation resistance of the EnDurCrete mortars compared to the reference mortars (see Figure 4) might therefore be related to the difference in the CH content in the uncarbonated mortar specimens. Since less CH is available in the EnDurCrete mortars compared to the reference mortars, these samples have a lower buffer capacity during carbonation (see, for example, [29,30,31]). Additionally, Revert et al. [21] have observed that low clinker binders develop a coarser porosity upon carbonation, enabling faster carbonation rates.

### 3.2. Moisture Transport

The mass changes of the thin plates during the moisture transport experiments are shown in Figure 6 for w/c ratio 0.6 and in Figure 7 for w/c ratio 0.5. The results in both figures are normalized to the initial mass of each plate. The standard sample deviation of mass changes found on a set of three specimens at any given time was between 5 and 10%. The rapid drop of mass at the beginning of stage 2 was probably due to surface drying of the freshly cut samples. The plates in the group “0%” continued to lose mass until equilibrium was reached between 96 and 98 wt% of the initial mass. When looking at the drying profiles of the “0%” series, we can see that the RefM-06 seemed to dry faster and to a larger extent compared to the EnM-06. The plates in the group “3%”, on the other hand, first decrease in mass but then start to increase in mass after ~3 days of exposure. At this point, the net mass gain due to carbonation (see Section 1) compensates for the mass loss due to drying. Sanjuán et al. [32] also reported an initial decrease in weight followed by mass gain, even though their specimens were stabilized at 60% RH before carbonation.

The only exception within the “3%” group, where the mass continues to decrease throughout stage 2, is EnM-06. The decrease or increase in mass in these experiments can suggest which of the two mechanisms (drying vs carbonation) has the strongest impact on the weight of the specimen. When CO_2_ is present, carbonation will cause weight gain due to the binding of CO_2_ in the reaction products. This weight gain will partially counteract the mass loss due to the simultaneous drying. According to Table 4, the EnM-06 mortar has a higher carbonation rate than the RefM-06; however, due to its lower clinker content, it had a lower CO_2_ binding capacity compared to the RefM-06. This might explain why the EnM-06 mortar did not gain any mass during simultaneous drying and carbonation, whereas the RefM-06 mortar did (stage 2 in Figure 6).

In stage 3, significant differences between groups “0%” and “3%” were observed. The uncarbonated plates in the former group absorbed significantly less water than the carbonated plates in the latter group (normalized to their initial weight). This indicates an increase in capillary porosity during carbonation compared to the uncarbonated samples. This assumption was confirmed by the porosity results shown in Figure 8a, where one can see that in the “-05” specimens porosity, increased by ~2% due to carbonation, which was not the case for the -06 samples. Although a decrease in porosity upon carbonation is commonly observed in Portland Cement-based binders [33,34], some studies reported an increase in total porosity in the case of cement blends utilizing supplementary cementitious materials [33,35,36]. The latter two studies also reported a coarsening of the pore structure. Justnes et al. [35] explained this by the fact that the reaction of C-S-H with a low Ca/Si ratio to CaCO_3_ causes a reduction in solid volume and, thereby, an increase in porosity [35]. The Ca/Si ratio of the C-S-H was not determined experimentally within this study. However, it can be assumed that, due to the high amount of slag present in both binders, the C-S-H phase formed shows a considerably lower Ca/Si ratio compared to C-S-H formed during the hydration of CEM I cement.

Stages 4 and 5 showed a gradual loss of mass. For the specimens in the “0%” group, the mass at the end of stage 5 (30% RH) was approximately the same as the mass at the end of stage 2 (60% RH). The normalized mass (to initial mass) of carbonated specimens was ~2% larger than the normalised mass of the uncarbonated specimens. 

The fact that the EnM-06 and RefM-06 were fully carbonated corresponds to the plateauing of the carbonated specimens during stage 2, as shown in Figure 6, confirming that these specimens reached equilibrium at those conditions. The corresponding curves for the carbonated EnM-05 and RefM-05 specimens, shown in Figure 7, did not plateau during stage 2, which reflects the fact that they had not yet been fully carbonated.

The degree of carbonation at the end of stage 2 was checked on one thin plate with the pH-indicator method. The plates were considered fully carbonated if the depth of carbonation was equal to half of the plate thickness, that is, 10 mm. The results for the plates in groups “3%” and “0%” are shown in Figure 8b. The plates in the “3%” group and with a w/c ratio of 0.6 (EnM-06 and RefM-06) carbonated completely, while those with a w/c ratio of 0.5 carbonated slightly less than half the depth, in the case of EnM-05, and about 5 mm, in the case of RefM-05. This is an undesirable outcome, since the numerical model calibration is to be performed on a fully carbonated material, on one hand, and on a completely uncarbonated material, on the other. The results show that the reduced CO_2_ levels in the conditioning chamber were able to limit the carbonation depth of the “0%” specimens. No carbonation depth was measured on the “0%” EnM-05 and RefM-05 specimens.

## 4. Conclusions

An experimental program was carried out to collect data for the verification of a numerical model by studying (1) the rate of carbonation, and (2) the impact of carbonation on moisture transport. For the verification of the rate of carbonation model, thick mortar blocks were exposed to accelerated carbonation conditions and the depth of carbonation was measured visually on split samples sprayed with a pH indicator and on profile-ground specimens using the TGA method. The impact of carbonation on moisture transport was monitored by measuring the mass of thin mortar plates subjected to a specific conditioning protocol.

The results obtained in this study indicate that carbonation depth is affected by the amount of clinker in the cement and by the w/c ratio. The lower the amount of clinker and/or the higher the w/c ratio, the higher the carbonation rate. Carbonation rates were found to be ~1.0 mm day^−0.5^ for the EnM-06 and ~0.6 mm day^−0.5^ for the Ref-06, while for the EnM-05 and RefM-05, the carbonation rates were ~0.7 and ~0.2 mm day^−0.5^ when exposed to accelerated carbonation in 1% CO_2_ atmosphere. The results of both methods, namely the visual and the TGA method, are in fairly good agreement. Additionally, the TGA showed that the initial CH content was ~1.5 wt% in the EnM-06, as opposed to ~3.0 wt% in the RefM-06. Based on the results of this study, the lower initial CH content of the EnDurCrete specimens might have been the reason for the higher carbonation rate.

The moisture transport experiments showed that carbonation increases the total porosity, which can facilitate the transport of deleterious substances, and therefore can promote other deterioration processes in addition to the corrosion of reinforcement. The results obtained on the EnM-06 showed that during carbonation at constant RH EnM-06 performed differently compared to other mortars. Unlike the results obtained in the carbonation rate experiment, during exposure to the CO_2_-rich environment, its mass continued to drop, while it exhibited the highest carbonation rate. The decrease or increase in mass shows which of the two mechanisms (drying vs carbonation) has the strongest impact. As an improvement to the experiment protocol, we recommend conditioning specimens at 60% RH after stage 1 until a constant mass is achieved, and then exposing them to carbonation.

The experimental results presented here will be used for the verification of mechanistic/generic models developed within the EnDurCrete project [5,11], which can be applied to novel cements with varying compositions. These models aim to improve the prediction of concrete durability in a CO_2_-rich environment and, therefore, reduce maintenance-related costs, particularly in the public infrastructure sector.

## Figures and Tables

**Figure 1 materials-14-06253-f001:**
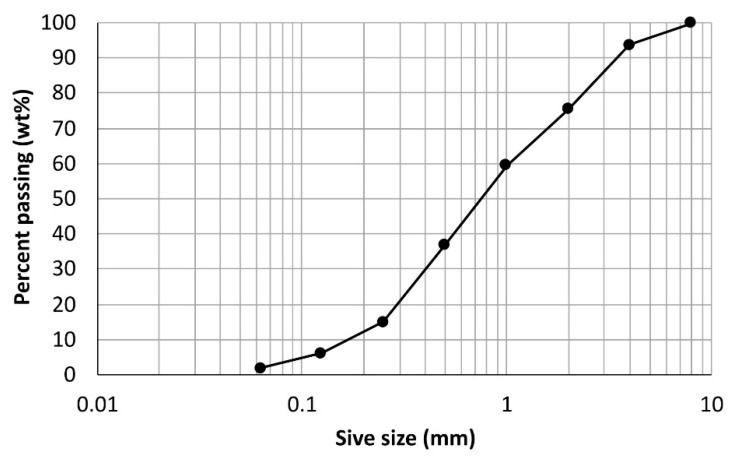
Particle size distribution of aggregate.

**Figure 2 materials-14-06253-f002:**
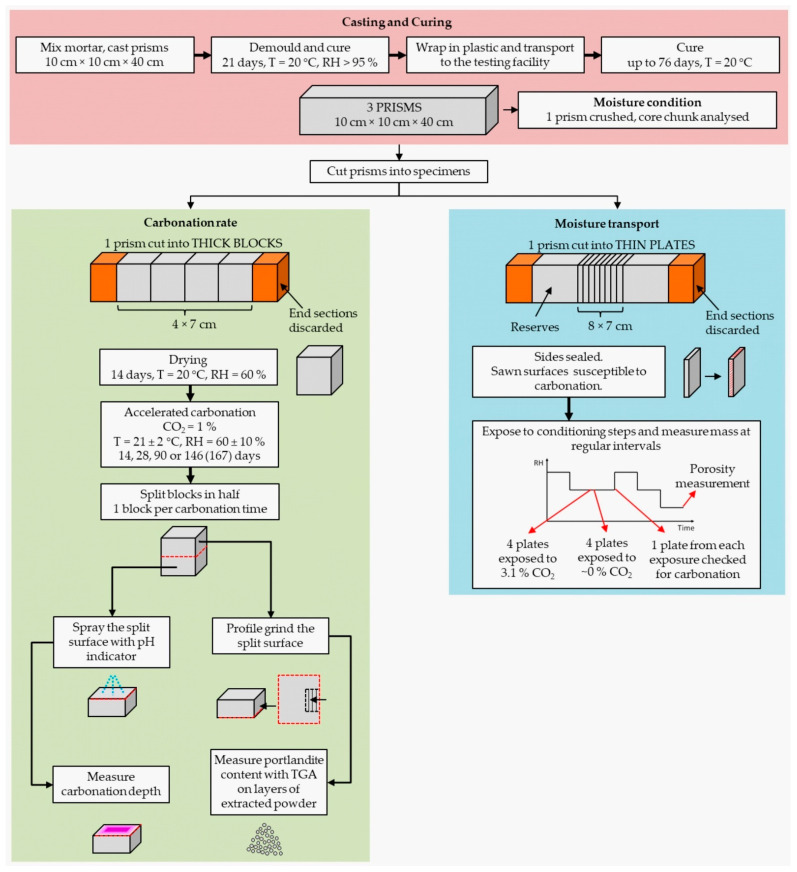
Flowchart showing the preparation, conditioning and testing of specimens. A detailed conditioning diagram for the moisture transport experiment is given in Figure 3.

**Figure 3 materials-14-06253-f003:**
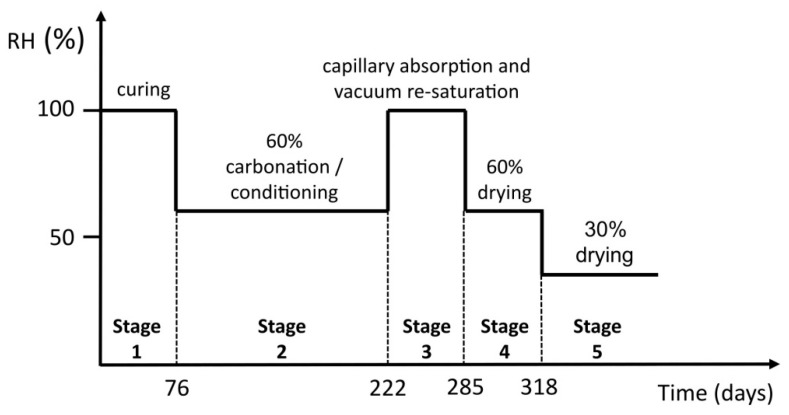
Stages of the moisture transport experiment carried out on the thin plates.

**Figure 4 materials-14-06253-f004:**
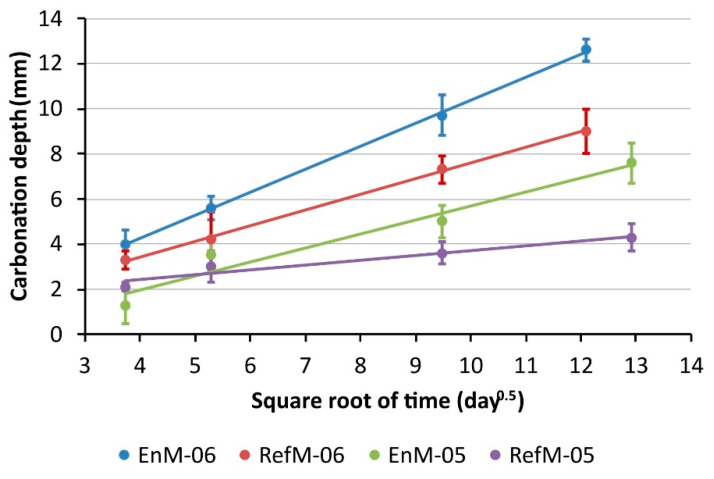
Carbonation depth, measured visually after spraying the split samples with a pH indicator, plotted against the square root of time. “EnM” and “RefM” stand for EnDurCrete and the reference mortar respectively, while the extensions “-06” and “-05” refer to water/cement ratios of 0.6 and 0.5. The error bars indicate the variation in the carbonation depth - measured in 10 pointson one sample for each mortar and exposure time.

**Figure 5 materials-14-06253-f005:**
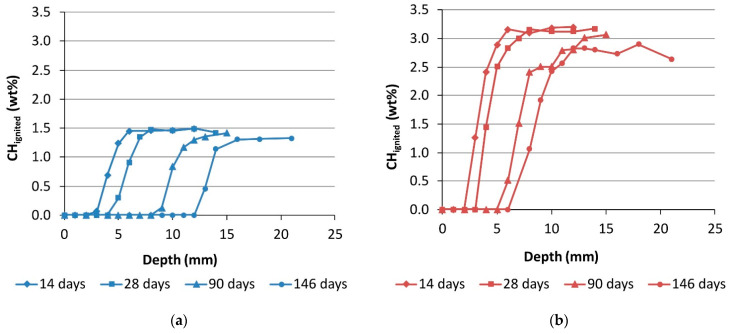
CH content measured with thermogravimetric analysis (TGA) after 14, 28, 90 and 146 days of exposure to accelerated carbonation for (**a**) EnDurCrete mortar (EnM-06); and (**b**) the reference mortar (RefM-06), Both made with a w/c ratio of 0.6.

**Figure 6 materials-14-06253-f006:**
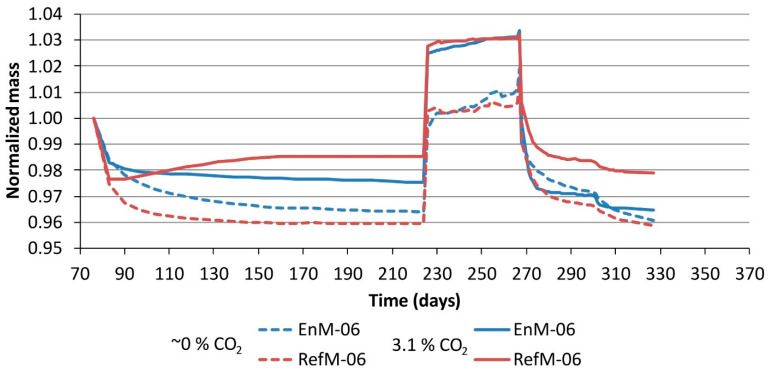
Mass changes during the moisture transport experiment for thin plates with a water/cement ratio of 0.6. In stage 2, group “3%” was exposed to carbonation in a 3.1% CO_2_ environment and group “0%” to conditioning in a ~0% CO_2_ environment. “EnM” and “RefM” stand for EnDurCrete and reference mortar respectively.

**Figure 7 materials-14-06253-f007:**
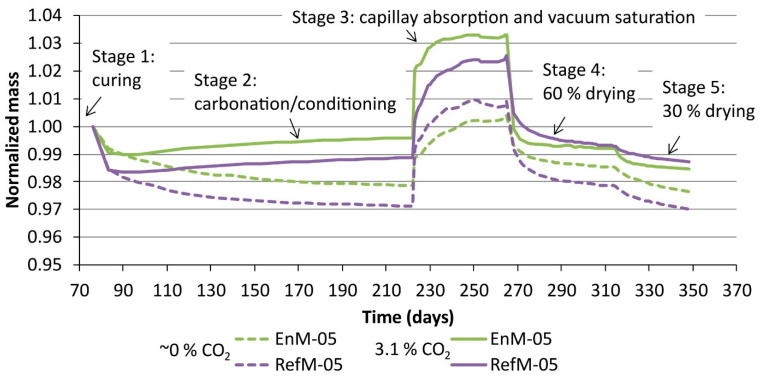
Mass changes during the moisture transport experiment for thin plates with a water/cement ratio of 0.5. In stage 2, group “3%” was exposed to carbonation in a 3.1% CO_2_ environment and group “0%” to conditioning in a ~0% CO_2_ environment. “EnM” and “RefM” stand for EnDurCrete and the reference mortar, respectively.

**Figure 8 materials-14-06253-f008:**
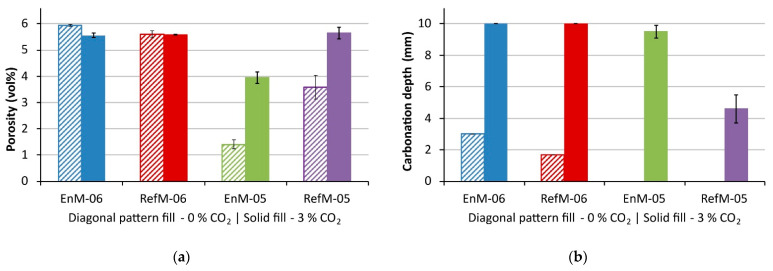
Total porosity (**a**) measured according to SIA 262-1 [23]; and carbonation depth (**b**) of conditioned specimens after stage 5. Specimens in the group “3%” were exposed to a 3.1% CO_2_ atmosphere, while specimens in the group “0%” were exposed to the atmosphere with reduced CO_2_ content. “EnM” and “RefM” stand for EnDurCrete and reference mortar, respectively, while extensions “-06” and “-05” refer to water/cement ratios of 0.6 and 0.5. The error bars indicate the variation in values measured on the three specimens.

**Table 1 materials-14-06253-t001:** Mortar mix design, stating the mass of components needed for 1 m^3^ of mortar. EnM and RefM stand for mortar made with EnDurCrete and reference cement respectively, while the extensions -06 and -05 denote w/c ratios.

Components		Mass of Component per 1 m^3^ of Mortar (kg)			
		EnM-06	RefM-06	EnM-05	RefM-05
Cement	CEM II/C (S-LL)	487	/	552	/
	CEM II/A-S	/	487	/	552
Aggregate	Sand 0/4	1524	1524	1498	1496
Admixture	Superplasticizer	1.5	1.8	3.2	3.8
Water		292	292	264	263

**Table 2 materials-14-06253-t002:** Properties of cement. The composition of cement is given as wt% of CEM I 52.5 R, which contains a minimum of 95 wt% of clinker (K), ground granulated blast-furnace slag (S) and limestone (LL). The chemical composition is given in Table A1 of the Appendix.

Property		CEM II/C (S-LL)	CEM II/A-S
Composition (wt%)	CEM I 52.5 R (K)	50	83
	S	40	13
	LL	10	4
Density (g cm^−3^)		2.98	3.09
Specific surface area–Blaine (cm^2^ g^−1^)		5210	3720
28-day compressive strength (MPa)		62.8	59.7

**Table 3 materials-14-06253-t003:** Physical properties of aggregate.

Property	Sand 0/4
Oven-dry density (kg m^−3^)	2730
Saturated surface-dry density (kg m^−3^)	2765
Apparent density (kg m^−3^)	2830
Absorption (%)	1.3

**Table 4 materials-14-06253-t004:** Results of regression analysis based on the measured values of carbonation depth at four different time intervals. As per (Equation (1)), parameter *K*_AC_ is the carbonation rate and parameter *a* is a constant representing the y-axis intercept, while R^2^ is the coefficient of determination indicating the quality of fit. “EnM” and “RefM” stand for EnDurCrete and the reference mortar respectively, while the extensions “-06” and “-05” refer to water/cement ratios of 0.6 and 0.5.

Parameter	EnM-06	RefM-06	EnM-05	RefM-05
*K*_AC_ (mm day^−0.5^)	1.03	0.64	0.66	0.23
*a* (mm)	0.15	1.22	−0.71	1.47
R^2^	0.98	0.92	0.88	0.79

## Data Availability

The data presented in this study are available on request from the corresponding author. The data are not publicly available due to the confidentiality conditions of the grant agreement.

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
