# Peer review of "Experimental Carbonation Study for a Durability Assessment of Novel Cementitious Materials"

_materials, 2021, doi:10.3390/ma14216253_

Round 1
Reviewer 1 Report
This paper provides an interesting accelerated carbonation test method for cementitious materials. However, the presentation of the results must be improved. Grammar and spell check is necessary for the manuscript.
- Please provide the details of the chemical composition and physical properties of the binder and aggregate.
- Please provide the details of the TGA test progress.
- Please provide the details of the test to determine the porosity of sample.
- Figure 2 should be improved to clearly show all the information.
- Please provide the TGA/DTG results and analysis of CH content based on the test results.
Author Response
Dear reviewer,
Thank you for checking our manuscript and providing us with guidelines for improvement. The summary of changes based on your recommendations is summarized below. Grammar and spelling were checked with Grammarly.
We kindly ask you to note, that this paper is submitted for the special issue of the journal featuring the contributions presented at the Conference on Construction Materials for a Sustainable Future (CoMS) 2020/21. At the conference, a page limit applied and for this reason, some important information had to be omitted. While in this revised version of the manuscript, the missing information is included, we attempted to preserve the essence of the paper presented at the conference. As advised by the journal editors, the following caveat was added just before the Introduction:
ll. 33-34: “This paper was presented at the 2nd International Conference on Construction Materials for a Sustainable Future, CoMS 2020/21.”
Please find our responses in the attached pdf file.
Kind regards,
Lucija

Reviewer 2 Report
The paper "Experimental carbonation study for durability assessment of novel cementitious materials" is interesting and presents advances related to the carbonation of cementitious materials, the authors should correct some points.
a) There are generally few references for a manuscript of this size and importance, which makes it difficult to discuss the results and present the state of the art, which is very deficient in this manuscript, I suggest the reading and introduction of some papers: 10.1016/j. conbuildmat.2017.10.111; 10.1007/s41024-020-00098-8; 10.1016/j.jmrt.2020.03.122; 10.1016/j.jobe.2021.102506.
b) Add an experimental flowchart in the methodology section;
c) The results need to be better discussed, use the manuscripts suggested by the reviewer and draw a comparative path in this research;
d) The conclusion is too long and not very objective for the readers.
Author Response
Dear reviewer,
Thank you for checking our manuscript and providing us with guidelines for improvement. The summary of changes based on your recommendations is summarized below.
We kindly ask you to note, that this paper is submitted for the special issue of the journal featuring the contributions presented at the Conference on Construction Materials for a Sustainable Future (CoMS) 2020/21. At the conference, a page limit applied and for this reason, some important information had to be omitted. While in this revised version of the manuscript, the missing information is included, we attempted to preserve the essence of the paper presented at the conference. As advised by the journal editors, the following caveat was added just before the Introduction:
ll. 33-34: “This paper was presented at the 2nd International Conference on Construction Materials for a Sustainable Future, CoMS 2020/21.”
Please find responses to your comments in the attached file.
Kind regards,
Lucija

Reviewer 3 Report
This paper includes a series of tests to characterize the carbonation of OPC concrete. In general, it's a very interesting paper, but there are some problems which the authors should look into.
- The irrelevant contents should not be included in the paper. The whole paper has nothing to do with any carbonation model. You cannot highlight what you are going to do next.
- The carbonation has been well studied by previous studies. The carbonation can generally decrease the pore volume and increase the strength, which is inconsistent with the results in the study.
Author Response

(The authors gave the same response as above.)

Reviewer 4 Report
The article presents the results of research on modified mortars containing a new type of CEM II / C cement, in conditions of accelerated carbonation and moisture transport, in order to verify the developed numerical model. However, the article does not present a numerical model. Moreover, the authors of the research do not show the results of the compressive strength tests of the samples, which would allow the proper evaluation of the tested mortars. There are a number of uncertainties and doubts in the work, which are presented below. This requires an explanation by the authors. The article is not eligible for publication in Materials as it stands. General remarks
- Why do the authors write in the article about the verification of a numerical model that they do not present?
- The paper specifies the degree of carbonation and the transport of moisture, completely ignoring the basic parameter of compressive strength. Failure to assess compressive strength significantly affects the validity of the tests carried out. Please explain why?
- Please explain what method was used to determine the steps for the moisture transport test?
- The results of the study did not present mean values and standard deviation for carbonation rate and moisture transport.
The article is not well written.
Specific remarks
- 24 line: Should be: mm / day-0.5
- 105 line: How many mortar prisms have been completed in total?
- 124 line Should be: "... 77 cm × 10 cm ..."
- 148 line: On what basis was this assumption made?
- 176 line: Why is the logarithmic scale not used on the abscissa?
- 189 line: Fig. 2: Describe in the text what the equations mean and whether R2 is satisfactory.
- 247 line: Fig.6: On the ordinate it should read "Carbonation…".
- 250 line: "in vačues" should be changed.
- 273 line: Shoull be: "4.Conclusions".
I recommend an in-depth review of the manuscript, including comments, to make it an article suitable for publication in the Materials.
Author Response
Dear reviewer,
Thank you for checking our manuscript and providing us with guidelines for improvement. The summary of changes based on your recommendations is summarized below.
We kindly ask you to note, that this paper is submitted for the special issue of the journal featuring the contributions presented at the Conference on Construction Materials for a Sustainable Future (CoMS) 2020/21. At the conference, a page limit applied and for this reason, some important information had to be omitted. While in this revised version of the manuscript, the missing information is included, we attempted to preserve the essence of the paper presented at the conference. As advised by the journal editors, the following caveat was added just before the Introduction:
ll. 33-34: “This paper was presented at the 2nd International Conference on Construction Materials for a Sustainable Future, CoMS 2020/21.”
Please find responses to your comments in the attached pdf file.
Kind regards,
Lucija

Round 2
Reviewer 1 Report
The authors provided sufficient information in the revised version to address all my comments in the previous version. Now this paper can be published.
Reviewer 2 Report
The manuscript may be accepted for publication in its current form.
Reviewer 3 Report
This paper has been revised based on the comments and should be ready for publication.